# Competition between antagonistic complement factors for a single protein on *N. meningitidis* rules disease susceptibility

**Joseph JE Caesar[1†], Hayley Lavender[1†], Philip N Ward[1†], Rachel M Exley[1], Jack Eaton[1], Emily Chittock[1], Talat H Malik[2], Elena Goiecoechea De Jorge[2], Matthew C Pickering[2], Christoph M Tang[1]\*, Susan M Lea[1]\***

[1]Sir William Dunn School of Pathology, University of Oxford, Oxford, United Kingdom; [2]Centre for Complement and Inflammation Research, Department of Medicine, Imperial College, London, United Kingdom

**Abstract** Genome-wide association studies have found variation within the complement factor H gene family links to host susceptibility to meningococcal disease caused by infection with *Neisseria meningitidis* (*Davila et al., 2010*). Mechanistic insights have been challenging since variation within this locus is complex and biological roles of the factor H-related proteins, unlike factor H, are incompletely understood. *N. meningitidis* subverts immune responses by hijacking a host-immune regulator, complement factor H (CFH), to the bacterial surface (*Schneider et al., 2006*; *Madico et al., 2007*; *Schneider et al., 2009*). We demonstrate that complement factor-H related 3 (CFHR3) promotes immune activation by acting as an antagonist of CFH. Conserved sequences between CFH and CFHR3 mean that the bacterium cannot sufficiently distinguish between these two serum proteins to allow it to hijack the regulator alone. The level of protection from complement attack achieved by circulating *N. meningitidis* therefore depends on the relative levels of CFH and CFHR3 in serum. These data may explain the association between genetic variation in both CFH and CFHR3 and susceptibility to meningococcal disease.

\*For correspondence: christoph. tang@path.ox.ac.uk (CMT); susan.lea@path.ox.ac.uk (SML)

†These authors contributed equally to this work

Competing interests: The authors declare that no competing interests exist.

## Introduction

*Neisseria meningitidis* is an important cause of rapidly progressive septicaemia and meningitis in children and young adults (*Stephens et al., 2007*); case fatality rates for bacteraemic disease remain at around 10%, and a significant proportion of survivors are left with long-term sequelae (*Vyse et al., 2013*). For most individuals, however, this human-specific bacterium is primarily a non-pathogenic commensal of the nasopharynx, and up to 40% of the population are healthy carriers (*Caugant and Maiden, 2009*). The mechanisms underlying human genetic influences that govern the development of invasive disease or asymptomatic carriage are incompletely understood. It is known, however, that complement activation is critical for protection against disease, evident from the susceptibility of individuals with genetic deficiency of either the alternative complement pathway (AP) or terminal pathway activation, and among those receiving therapy that prevents terminal pathway activation (*Schneider et al., 2007*; *McKeage, 2011*). Amongst several strategies that promote complement evasion, *N. meningitidis* recruits the human negative complement regulator, complement factor H (CFH), to its surface by expressing factor H binding protein (fHbp) (*Schneider et al., 2006*, *2009*; *Madico et al., 2007*). fHbp can be divided into three variant groups (V1, V2, and V3), which have >85% sequence identity within the groups but only 60–70% similarity between groups (*Masignani et al., 2003*;

**eLife digest** Meningitis is a potentially life-threatening condition whereby the membranes that cover and protect the brain and spinal cord become inflamed. Often meningitis is caused by a viral or bacterial infection—such as infection by a bacterium called *Neisseria meningitidis*, also known as meningococcus. However, not everyone that comes into contact with this bacterium will develop meningitis; 40% of the population is thought to carry *N. meningitidis* at the back of the nasal cavity and yet show no signs of the disease.

It remains unclear why some people exposed to *N. meningitidis* develop meningitis while others do not; however recent research revealed that part of the immune system called the complement system plays a role in susceptibility to meningitis. The complement system is a collection of small proteins that work together to support the actions of the cells of the immune system. When activated, complement proteins trigger a cascade of events that helps to destroy the pathogen.

Several mechanisms exist to keep the complement proteins in check—for example, a protein called complement factor H (or CFH) protects host cells from being attacked by other complement proteins. *N. meningitidis* can undermine the complement system by expressing a protein that binds to CFH and firmly fixes CFH to its cell surface. While the CFH-binding protein helps explain why some people are unable to mount the appropriate immune response to infection by *N. meningitidis*, it does not explain why some carriers of the pathogen do not develop meningitis.

Now, Caesar et al. have examined a protein called CFH related-3 (or CFHR3), and discovered that CFHR3 competes with CFH for the binding protein on *N. meningitidis*. CFHR3 is structurally similar to CFH, but it is unable to regulate or silence the complement system. Caesar et al. explain that susceptibility to meningococcal disease is determined by how much CFH and how much CFHR3 each individual has, and that those with less CFHR3 will be more susceptible to *N. meningitidis*. An individual's genes will affect how much CFH and CFHR3 they have, while the genes of the bacterium can influence how strongly the CFH binding protein binds to either of these human proteins. Caesar et al. suggest that these two factors determine whether or not an individual will develop meningitis or simply carry the bacterium without any ill effects.

Caesar et al.'s findings highlight the different ways that people's genes can determine how they respond to an invading pathogen. The findings also suggest that it is important to consider variation in the levels of these complement proteins across a population when planning immunisation schedules.

*Brehony et al., 2009*). fHbps from all variant groups bind CFH with a $K_D$ in the low nanomolar range (*Johnson et al., 2012*). Therefore, it is significant that the only single nucleotide polymorphisms (SNPs) associated with meningococcal disease in a recent genome-wide association study (GWAS) were those among the *CFH–CFHR* locus. These included SNPs within *CFH* and the downstream gene, *CFHR3* (*Davila et al., 2010*). While CFH has been extensively studied (reviewed in *Makou et al. (2013)*), the role of the serum protein CFHR3 has not been unambiguously established (reviewed in *Jozsi and Meri (2014)*).

## Results and discussion

We recently demonstrated that three of the five CFHR proteins (i.e., CFHR1, 2, and 5) compete with CFH for binding to C3b on heterologous erythrocytes, thereby promoting complement activation (*Goicoechea de Jorge et al., 2013*). Given the association of CFHR3 with meningococcal disease (*Davila et al., 2010*), we investigated whether this protein and CFHR4 also act as CFH antagonists in this assay. We generated full-length CFHR3 and CFHR4B (which both consist of five complement control protein [CCP] domains, *Figure 1A*) and truncated versions of these proteins, and we examined their ability to influence complement activation by the AP, which is regulated by CFH (*Makou et al., 2013*). Erythrocyte haemolysis assays demonstrate that CFHR3 and CFHR4B are also CFH antagonists on this surface, with their two C-terminal domains enabling them to compete with CFH for binding sites on cell complement fragments such as C3b (*Figure 1B*, *Figure 1—figure supplement 1A,B*). Additional cell-surface recognition sites within other domains (*Figure 1A*) increase the activity of the full-length CFHR3 and CFHR4B as competitive antagonists of CFH (*Figure 1B*) at physiologically relevant concentrations of ~1 µM (*Fritsche et al., 2010*; *Hebecker and Jozsi, 2012*).

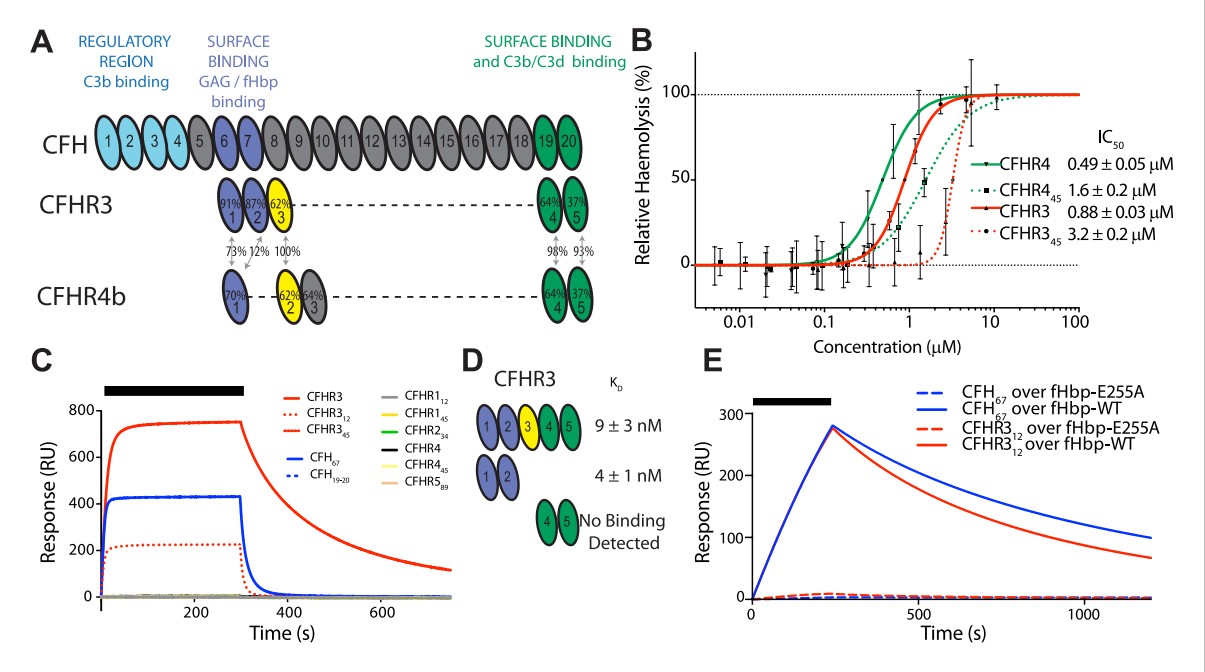

**Figure 1**. CFHR3 binds *N. meningitidis* fHbp at the same site as CFH and promotes complement activation. (**A**) Complement control protein (CCP) domains of CFH, CFHR3, and CFHR4 are shown with the sequence identity to CFH and between the CFHRs indicated. Key functional regions of CFH are noted. (**B**) CFH-dependent alternative pathway (AP) haemolytic assay (***Goicoechea de Jorge et al., 2013***). Using a CFH dose that reduced lysis of guinea pig erythrocytes to 50%, the addition of increasing concentrations of either full-length or C-terminal fragments of CFHR3 or CFHR4 resulted in a dose-dependent increase in lysis. The two N-terminal CCP domains from CFHR3 had no effect (not shown). (**C**) Surface plasmon resonance (SPR) was used to investigate interactions of CFHRs with fHbp. Full-length and fragments of CFH and CFHRs were injected (black bar) over surface bound fHbp$_{3.28}$. Only CFHR3$_{12}$, full-length CFHR3, or CFH$_{67}$ bind to fHbp. (**D**) Dissociation constants ($K_D$) measured by SPR (CFHR3$_{12}$) or microscale thermophoresis (CFHR3) indicated the fHbp binding site is within the two N-terminal domains of CFHR3. (**E**) SPR with 200 nM protein flowing over V3.28 fHbp demonstrates that the single point mutation E255A, which ablates CFH binding, also ablates CFHR$_{12}$ binding.

The following figure supplement is available for figure 1:

**Figure supplement 1**. Increased haemolysis on addition of CFHR3 is dependent on presence of CFH and interaction of CFHR3 full length and N-terminal fragment with V3.28 fHbp.

*N. meningitidis* fHbp recruits CFH by high-affinity interactions primarily with domain 6 of CFH (CFH$_6$) with some limited contact to CFH$_7$ (***Schneider et al., 2009***). Since the first domains of the CFHRs share between 40% to 90% identity with CFH$_6$ (***Figure 1A***, ***Figure 2***), we next investigated whether fHbp also bound the CFHRs. Our findings (***Figure 1C***) demonstrate that CFHR3 binds fHbp with a $K_D$ of ~3 nM, similar to affinities for CFH (***Schneider et al., 2009***; ***Johnson et al., 2012***); no significant binding was detected for the other CFHRs with fHbp. Of note, the first two domains of CFHR3 (CFHR3$_{12}$; domain 1 shares 91% identity with CFH$_6$, ***Figure 1A***) and the full-length protein bind fHbp with similar affinities (***Figure 1E***, ***Figure 1—figure supplement 1C,D***). Furthermore, a point mutation within fHbp (E255A) that reduces binding to CFH by more than three orders of magnitude (***Johnson et al., 2012***) similarly reduces CFHR3 binding (***Figure 1E***). The nature of the CFH and CFHR3 interactions with fHbp are therefore similar, with these complement factors occupying overlapping sites on fHbp, which is consistent with the high degree of conservation of their fHbp binding sites (***Figure 2***).

Next we investigated whether CFHR3 binds to the surface of *N. meningitidis*. We generated a CFHR3-specific monoclonal antibody (mAb) (***Figure 3—figure supplement 1***), designated HSL1. By flow cytometry we demonstrated that serum CFHR3 bound *N. meningitidis* M1239 (which expresses V3.28 fHbp) in a fHbp-dependent manner. No binding was detected following incubation of bacteria in sera from an individual homozygous for an allele in which there is combined deletion of the *CFHR1* and *CFHR3* genes (Δ*CFHR3/1*) (Δ*CFHR3/1*, ***Figure 3A,B,C***). We therefore tested the functional consequences of CFHR3 binding by comparing the effect of adding full length or domains of CFHR3

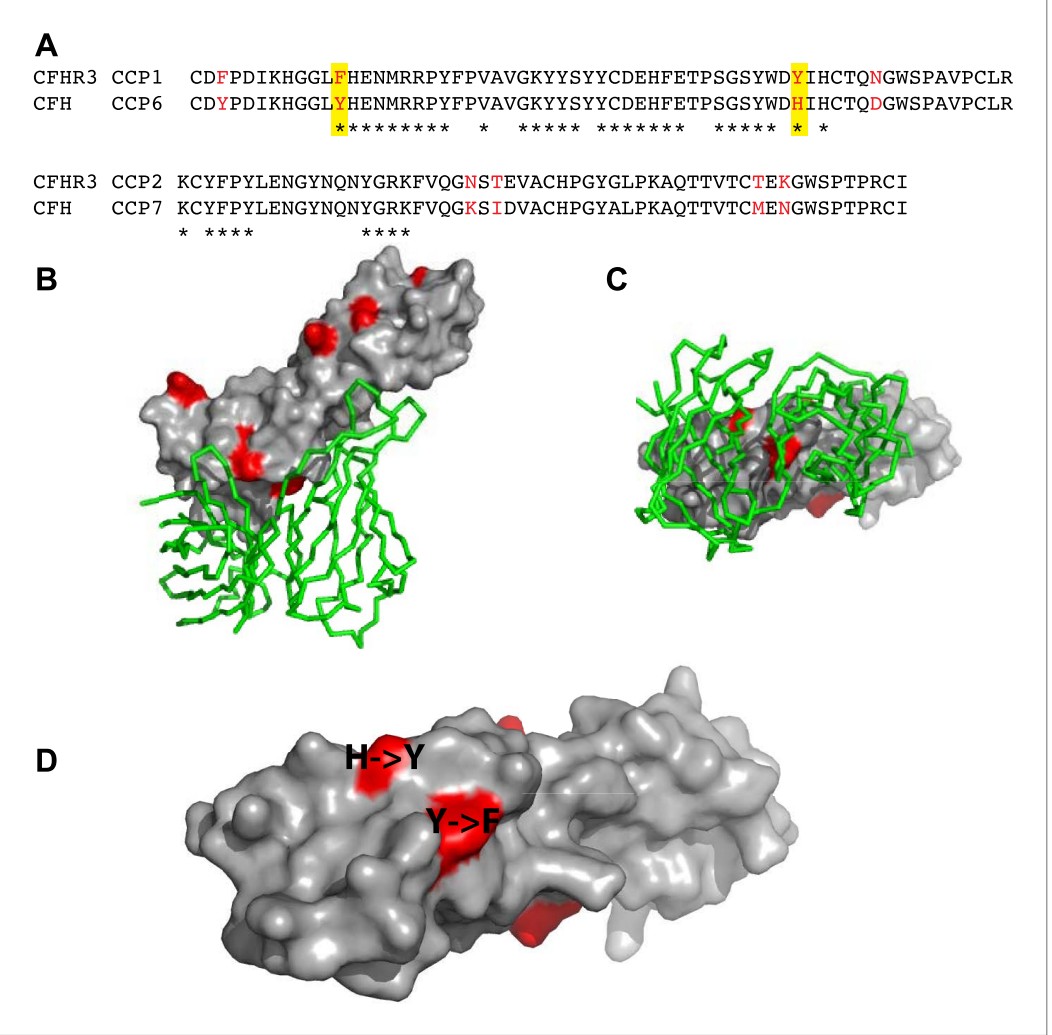

**Figure 2**. Amino acid differences between CFHR3 and CFH in the fHbp binding site. (**A**) Sequence alignment between CFHR3$_{12}$ and CFH$_{67}$. Differences are highlighted in red text, the two differences within the fHbp binding site are highlighted in yellow and residues contacting fHbp in the CFH-fHbpV3 structure (PDB ID 4AYM [***Johnson et al., 2012***]), chains A and D are indicated by an asterisk below the CFH sequence. (**B**) CFH$_{67}$ is shown as a grey surface with residues that differ in CFHR3 highlighted in red; fHbp is shown as a green ribbon. (**C**) Rotated 90° around x compared to the view in **B**. (**D**) Looking down onto the fHbp binding site on CFH the two, conservative, sequence differences between the CFH$_{67}$ and CFHR3$_{12}$ found in the binding site are indicated in red on the grey surface.

to *N. meningitidis* prior to incubation in normal human serum (NHS). Addition of full-length CFHR3 or CFHR3$_{12}$, which contains the fHbp binding site (***Figure 1C***) but has no CFH antagonist activity (***Figure 1B***), resulted in significantly reduced bacterial survival (CFHR3 51%; CFHR3$_{12}$ 70%; no protein or CFHR3$_{45}$ 83%, ***Figure 3D***); in contrast, CFHR3 had no detectable effect in the presence of heat-inactivated NHS (which lacks complement activity, ***Figure 3D***). Full-length CFHR3 has a more marked effect on bacterial survival than CFHR3$_{12}$, consistent with the ability of the full-length CHHR3 to bind to C3b deposited on the bacterial surface (via CFHR3$_{45}$) as well as competing with CFH for fHbp (via CFHR3$_{12}$). As CFHR3$_{45}$ does not bind fHbp (***Figure 1C***), its CFH antagonist activity is not effectively localized to the meningococcal surface, providing an explanation for why these domains have no effect on bacterial survival (***Figure 3D***). Similarly, addition of CFHR3 has no effect on serum survival of bacteria that do not express any CFH-binding protein (***Figure 3—figure supplement 2A***, *N. meningitidis* strain lacking fHbp expression; ***Figure 3—figure supplement 2B***, Escherichia *coli* strain with no expression of any CFH-binding protein).

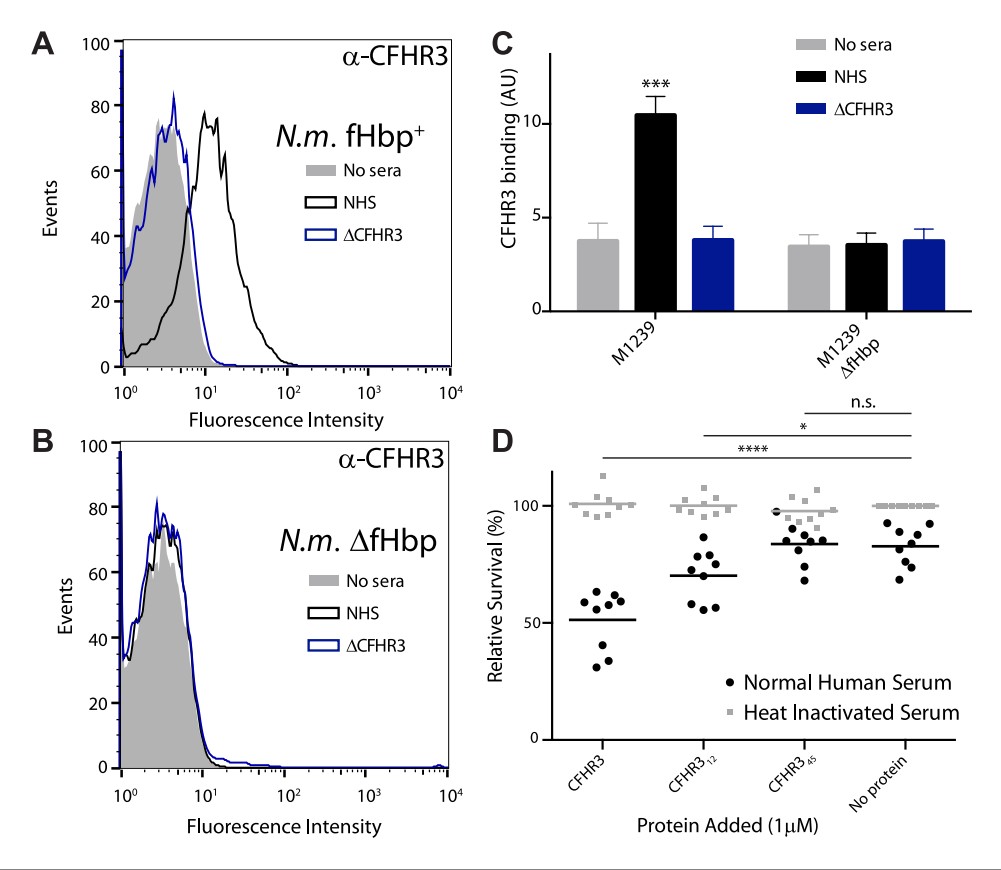

**Figure 3**. *N. meningitidis* binds CFHR3 on its surface in an fHbp-dependent manner promoting complement-mediated lysis. (**A**–**C**) Flow cytometry of *N. meningitidis* demonstrates that CFHR3 binds to fHbp on the bacterial surface. (**D**) Sensitivity of *N. meningitidis* strain M1239 to complement-mediated cell lysis after pre-incubation with 1 μM CFHR3, CFHR3$_{12}$, CFHR3$_{45}$, or no protein. Faded symbols are bacteria incubated in identical conditions except using heat-inactivated NHS instead of NHS. Data presented as percentage survival relative to bacteria incubated in heat-inactivated NHS without additional CFHR3. Line shows the mean of three independent biological experiments, each with three technical replicates; individual results indicated. Significance was calculated using a two-tailed unpaired *t* test comparing with values obtained with no additional protein: ****$p < 0.0001$; *$p < 0.05$; ns $p > 0.05$.

The following figure supplements are available for figure 3:

**Figure supplement 1**. Characterisation of the novel anti-CFHR3 antibody HSL1.

**Figure supplement 2**. The effect of CFHR312 on bacterial survival is dose and fHbp dependent.

The finding that CFHR3 binding impairs bacterial survival in NHS led us to consider whether fHbp can distinguish between the limited sequence differences between CFH and CFHR3 to generate specificity for CFH. We previously identified a series of alanine substitutions in fHbp that significantly reduce or ablate CFH binding (*Johnson et al., 2012*). Therefore, we measured the contribution of fHbp residues to interactions with CFHR3 to determine whether any amino acid changes altered the specificity of fHbp for the two complement factors (*Figure 4A*, *Figure 4—source data 1*). The results demonstrate that the majority of mutations that impair CFH binding (*Johnson et al., 2012*) also significantly reduce CFHR3 binding, so do not alter specificity for CFH; point mutations at only a few sites led to modest alterations in specificity at levels that are unlikely to be physiologically relevant given the relative affinities for CFHR3/CFH and their abundance in serum (CFHR3 1–1.6 μM [*Fritsche et al., 2010*; *Hebecker and Jozsi, 2012*]; CFH 1–4 μM [*Hakobyan et al., 2008*]). Interestingly the V1 fHbp used as the background for the mutations (V1.1) did reveal a degree of specificity for CFH over

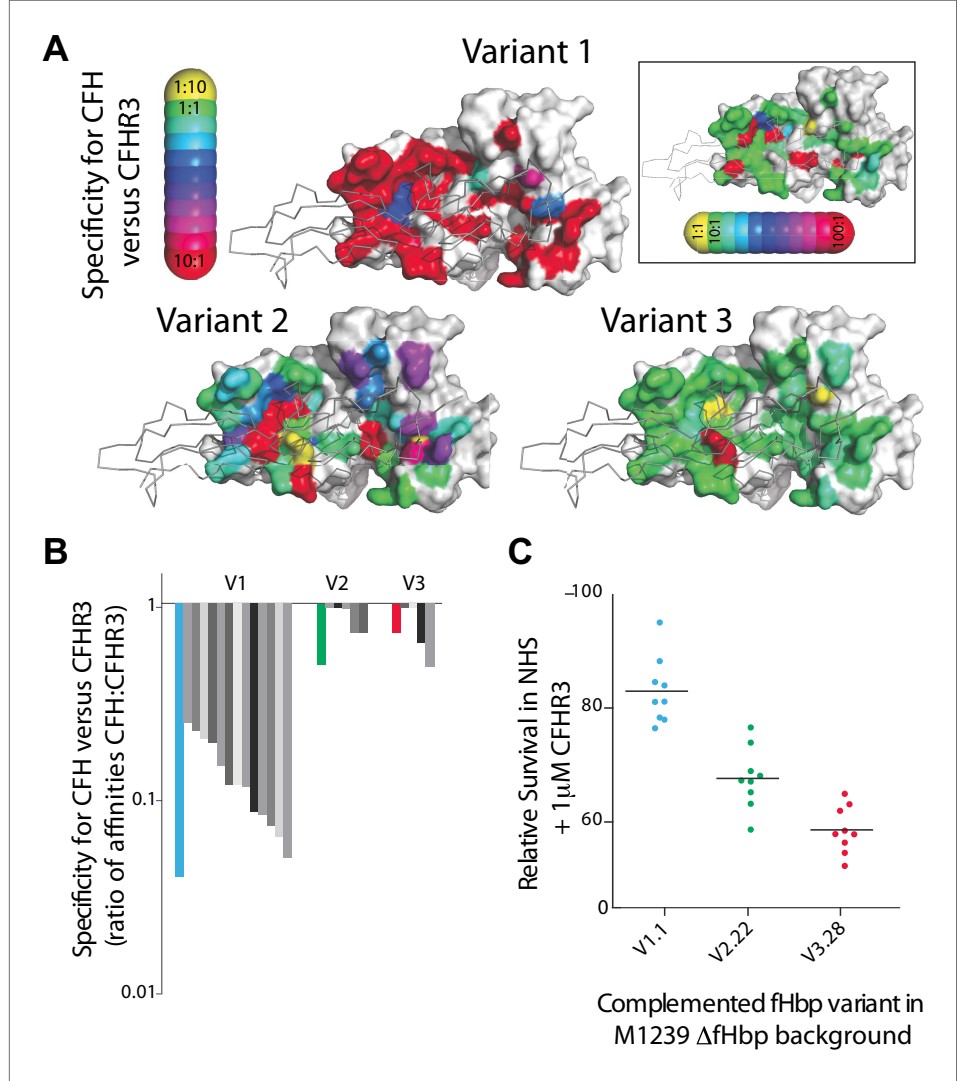

**Figure 4**. Key residues in fHbp that mediate binding to CFHR3. (**A**) SPR was used to determine $K_D$s for both $CFH_{67}$ and $CFHR3_{12}$ binding to a panel of fHbps bearing alanine substitutions within the $CFH_{67}$ binding site. ~50 mutations were screened in each of three fHbps (V1.1, 2.21 and 3.28 [**Johnson et al., 2012**]). Mutations altered $K_D$s for both $CFH_{67}$ and $CFHR3_{12}$ but some mutations led to some specificity for binding one or the other protein (see full data in Extended Data File 1). (**B**) $K_D$s were determined for a panel of fHbps from disease causing *N. meningitidis*. V1 fHbps have some ability to preferentially bind $CFH_{67}$ vs $CFHR3_{12}$, whilst V2 and V3 fHbps were less able to distinguish the molecules (full data in Extended Data File 2). (**C**) AP-mediated bacterial killing of isogenic strains expressing different fHbps in the presence of CFHR3. Data are coloured as in panel (**B**) and bacterial survival in serum is increased in the strain expressing the fHbp$_{V1.1}$ (that has the maximal specificity for CFH vs CFHR3).

The following source data is available for figure 4:

**Source data 1**. Excel spreadsheet with full details of $K_D$s for CFH and $CFHR3_{12}$ derived from 1:1 Langmuir fits of SPR data for alanine scanning mutants and natural variant fHbp sequences.

CFHR3 (in the order of ~20-fold) in contrast to V2 and V3 fHbps, suggesting that certain fHbps can discriminate between CFH and CFHR3.

To investigate this further, we measured binding of CFH and CFHR3 to 25 naturally occurring fHbps that are prevalent in disease isolates of *N. meningitidis* (**Figure 4B**, **Figure 4—source data 1**) (**Lucidarme et al., 2009**). This demonstrated that V2 and V3 fHbps displayed no selective affinity for CFH over CFHR3 (average selectivity being 1.8 ± 1.4 and 2.4 ± 1.0, respectively). In contrast, V1 fHbps

had lower affinities for CFHR3 compared to CFH (average selectivity 8.9 ± 4.6-fold), with 12 of the 14 V1 fHbp variants binding CFH 5- to 20-fold more tightly than CFHR3. Modeling the effect on the occupancy of fHbp with CFH indicates that certain V1 fHbps will be bound by CFH in the physiological situation regardless of CFHR3 levels (*Figure 5*). To examine whether this has any functional consequence, we examined complement AP killing of isogenic strains expressing fHbp sequences with different abilities to discriminate between CFH and CFHR3 (*Figure 4C*); AP killing was assessed to exclude any confounding influence of antibodies in sera preferentially recognizing different fHbps. Bacteria expressing V3.28 fHbp, which has identical affinities for CFH and CFHR3, were the most sensitive to AP killing, while the strain expressing the V1.1 fHbp sequence (which has ~20-fold tighter binding to CFH than CFHR3) was the least sensitive; bacteria with fHbps with intermediate CFH specificity displayed intermediate levels of protection. This suggests that, even though the most specific fHbps bind CFHR3 with a $K_D$ significantly below the physiological concentration (*Figure 4— source data 1*), the ability of fHbp to favour CFH binding promotes bacterial survival and offers a potential explanation for the prevalence of strains expressing V1 fHbp causing invasive disease (*Lucidarme et al., 2009*).

To define the basis for the effect of CFHR3 on bacterial survival, we next examined the impact of CFHR3 on levels of CFH bound to the bacterial surface. *N. meningitidis* incubated in the *ΔCFHR3/1* sera acquire high levels of CFH on the bacterial surface (*Figure 6A,B,C*). However, addition of

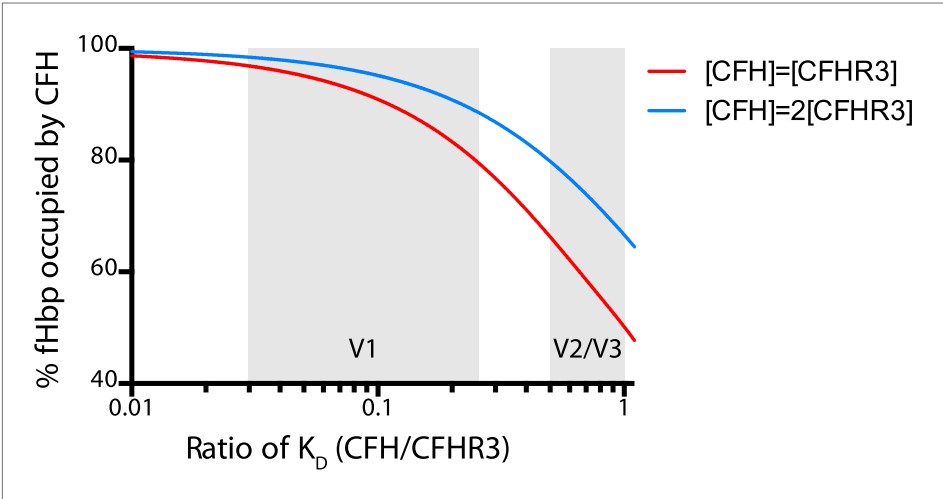

**Figure 5**. Modelling the occupancy of bacterial cell surface fHbp assuming different levels of specificity for CFH over CFHR3. Each molecule of fHbp on the bacterial surface can bind with one molecule of CFHR3 or one molecule of CFH. Assuming that the number of molecules of either of these is significantly greater than the number of molecules of fHbp then the percentage of bacterial surface fHbp molecules binding CFH at equilibrium will be determined by: the relative amounts of both present in the individual; and the affinities of that fHbp sequence for CFH vs CFHR3 (essentially a model of competitive antagonism, see equation below). Natural fHbp sequences are relatively constant in their $K_D$ for CFH ($K_D$ ~3 nM [*Johnson et al., 2012*] and this work), therefore we modelled the effect of the $K_D$ for CFHR3 varying between 3 nM and 300 nM against this constant $K_D$ for CFH under two conditions. In red we assume that both CFH and CFHR3 are present in serum at 1.5 μM, whilst in blue we model the effect if the concentration of CFH is twice that of CFHR3 (i.e., 3 μM vs 1.5 μM). This range of relative concentrations covers those generally reported for CFH and CFHR3 (except in the case of *ΔCFHR3/1* individuals where the concentration of CFHR3 will be zero and 100% of the fHbp will bind CFH). The grey shaded areas indicate the regions of the x axis into which the families of natural variant fHbp sequences studied in *Figure 3* fall. The equation used to generate these curves is:

$$Occupancy\,CFH = \frac{\dfrac{[CFH]}{K_D(CFH)}}{\dfrac{[CFH]}{K_D(CFH)} + \dfrac{[CFHR3]}{K_D(CFHR3)} + 1} * 100.$$

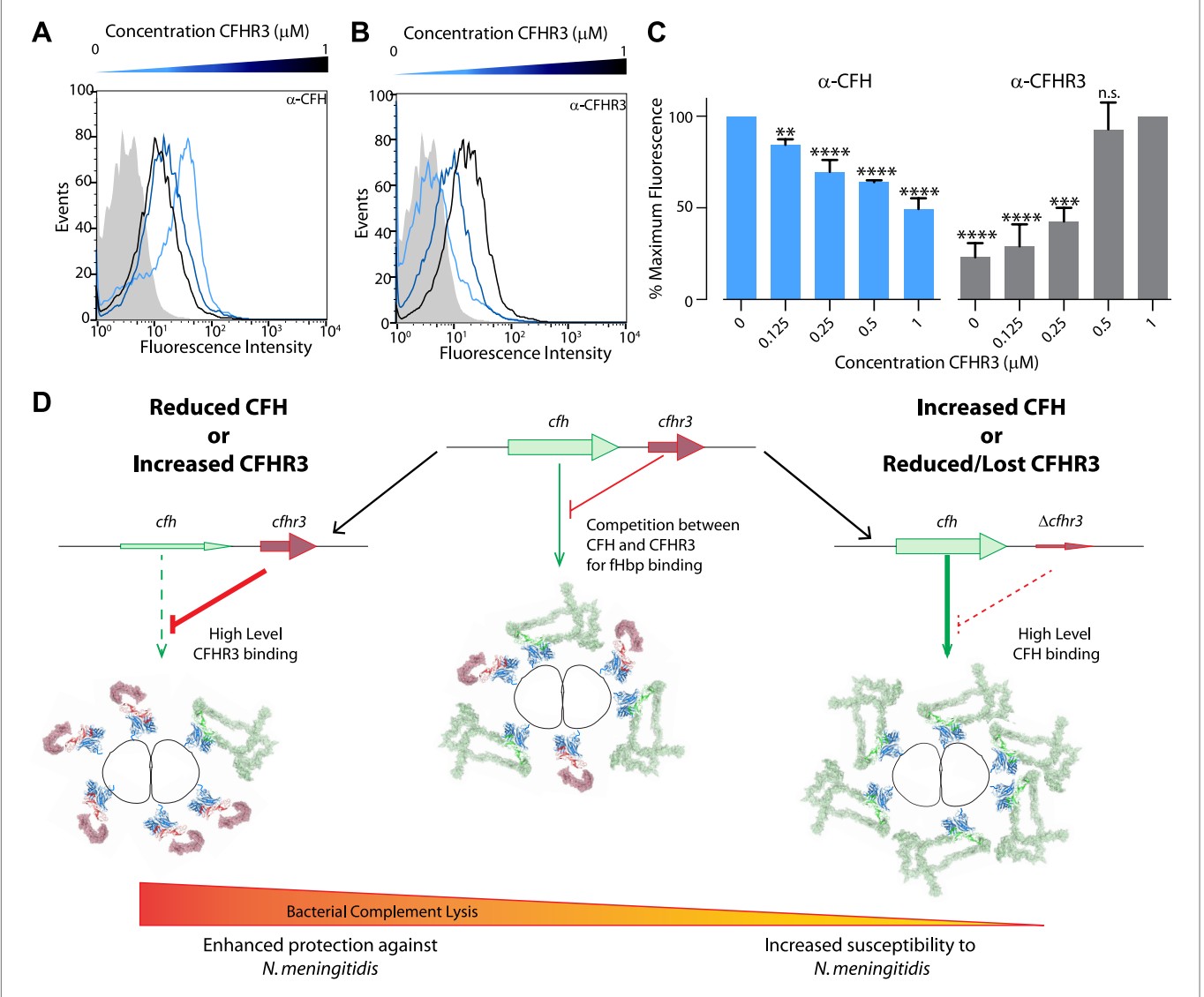

**Figure 6**. Susceptibility to *N. meningitidis* will be altered by the circulating levels of CFH and CFHR3 that compete for binding to fHbp. (**A–C**) Flow cytometry demonstrates that addition of increasing amounts of CFHR3 (from 0 μM [light blue] to 1 μM [black]) to sera from a *Δ*CFHR3–1 individual leads to decreasing levels of CFH on the bacterial surface (**A** and **C**) and increasing levels of CFHR3 (**B** and **C**). (**D**) Model of how differing levels of CFHR3 and CFH in an individual combine to alter the amount of CFH hijacked to the bacterial surface, hence the level of complement-mediated bacterial killing and an individual's susceptibility to meningococcal disease.

recombinant CFHR3 reduces the amount of bound CFH in a dose-dependent manner, demonstrating antagonism between CFH and CFHR3 for binding to the bacterial surface. When the amount of recombinant CFHR3 is added at approximately equimolar levels to those in NHS, the level of CFH bound on the bacterial surface is only half that seen in the CFHR3-deficient sera (1–1.6 μM [*Fritsche et al., 2010*]).

In summary, our work demonstrates that susceptibility to an important infectious disease is governed by the competition between two complement factors with opposing effects (CFH and CFHR3) for overlapping binding sites on a single pathogen molecule. Our data predict that the risk of developing meningococcal disease will be governed by the relative abundance of CFH and CFHR3 in individuals and the affinity of these molecules for the specific fHbp on the bacterial surface (*Figure 6D*). It is notable that the *ΔCFHR3/1* allele has a high prevalence in sub-Saharan Africa (*Holmes et al., 2013*) where there are high rates of epidemic disease. Our mechanism is consistent with this association,

since this allele would enhance the interaction between CFH and fHbp and favour pathogen survival. This is particularly true as the $\Delta CFHR3/1$ deletion is associated with increased plasma CFH levels, most likely due to the loss of a regulatory locus (**Ansari et al., 2013**; **Zhu et al., 2014**). However, host polymorphisms in CFHR3 might have less impact on strains of bacteria that express fHbp that more selectively bind CFH. Therefore, susceptibility to meningococcal disease will be influenced by polymorphisms and copy number variation affecting two host complement factors and by the relative affinity of fHbp on an infecting strain for CFH and CFHR3. This illustrates why genetic association studies in infectious diseases are likely to be technically challenging since these will be influenced not only by sequence and structural variation within the host but also by genetic variation in pathogen.

## Materials and methods

### Cloning and mutagenesis for protein expression

Sequences encoding $CFHR1_{45}$, $CFHR3_{12}$, $CFHR3_{45}$ and $CFHR4_{45}$ were amplified using primer pairs CFHR1-4-Forward(5′-GAAGGAGATATACCATGACGGGAAAATGTGGGCC-3′)/CFHR1-5-Reverse(5′-GGCCCACATTTTCCCGTCATGGTATATCTCCTTC-3′), CFHR3-1-Forward(5′-GATATACATATGAAACCTTGTGATTTTCCAGACATTAAAC-3′)/CFHR3-2-Reverse(5′-GGATCCTCGAGCTAACGGATGCATCTGGGAGTAG-3′), CFHR3-4-Forward(5′-GGCAGCCATATGTCAGAAAAGTGTGGGCCTCCTC-3′)/CFHR3-5-Reverse(5′- CGGATCCTCGAGTTATTCGCATCTGGGGTATTCCACTATC-3′) and CFHR3-4-Forward/CFHR4-5-Reverse(5′- CGGATCCTCGAGTTATTCGCATCTGGGGTATTCCACTATG-3′), respectively. The sequence encoding $CFHR5_{89}$ was amplified using primer pair CFHR5-8-Forward(5′-CCACATATGGCATATTGTGGGCCCCCTCCATC-3′)/CFHR5-9-Reverse(5′-GGATCCTCGAGTCATTCACATATAGGATATTCAAATTTC-3′). Inserts were cloned between the NdeI and XhoI sites of a modified version of vector pET-15b (Novagen, Merck Millipore, Billerica, MA), which has the NcoI site replaced with an NdeI site.

A vector encoding $FH_{19-20}$ was generated from that encoding $CFHR1_{45}$ using site-directed mutagenesis with primers L183S-Forward(5′-GGACAGCCAAACAGAAGCTTTATTCGAGAACAGGTGAA-3′)/L183S-Reverse(5′-TTCACCTGTTCTCGAATAAAGCTTCTGTTTGGCTGTCC-3′) and A189V-Forward(5′-TTTGAGAACAGGTGAATCAGTTGAATTTGTGTGTAAACGGG-3′)/A189V-Reverse(5′-CCCGTTTACACACAAATTCAACTGATTCACCTGTTCTCAAA-3′).

The gene encoding full-length CFHR3 was amplified using primers encoding a histidine (His) tag (CFHR3-Forward 5′-ATTCGCGGCCGCCCCCACCATGTTGTTACTAATCAAT-3′ and CFHR3-Reverse 5′-CACCATCACCATCACCATTAGCTCGAGGAA-3′) and cloned into a modified version of the pCAGGS plasmid (**Shimizu et al., 1999**) bearing the CMV-EI enhancer, the chicken β-actin promoter, and intron 1 of the simian virus 40 poly(A) signal.

The gene encoding full-length CFHR4 was amplified using primer pair CFHR4-Forward(5′- GGACTGTCTAGAGACTCAGATCCCCATCCGCTCAAGCAGGCCACCATGTTGTTACTAATCAATGTCATTCTGACC-3′)/CFHR4-Reverse(5′ GTGTCGGGTCGACCCTTCGCATCTGGGGTATTCCAC-3′) prior to ligation into vector pEF-BOS (**Mizushima and Nagata, 1990**) between the XbaI and SalI sites giving a C-terminal hexahistidine tag.

### Recombinant protein expression and purification

Recombinant His-tagged fHbp was prepared from a selection of natural variant fHbps and alanine substitution mutants of fHbp V1, V2, and V3 as described previously (**Johnson et al., 2012**). All three variant types were represented within 27 natural fHbp variant sequences, numbered according to an allele ID code as detailed on the Neisseria Multi Locus Sequence Typing website (http://pubmlst.org/neisseria/) developed by Keith Jolley and sited at the University of Oxford (**Jolley and Maiden, 2010**).

$CFH_{67}$ was prepared as previously described (**Prosser et al., 2007**). $CFH_{19-20}$, $CFHR1_{45}$, $CFHR3_{45}$, $CFHR4_{45}$, and $CFHR5_{89}$ were expressed in *E. coli* and refolded from inclusion bodies prior to size exclusion chromatography (Superdex S-75, GE Healthcare, Little Chalfont, UK, buffer 50 mM Tris pH 7.5, 150 mM NaCl) using the same method previously described for $CFHR2_{45}$ (**Goicoechea de Jorge et al., 2013**). $CFHR1_{12}$ was expressed in *Kluyveromyces lactis* as previously described (**Goicoechea de Jorge et al., 2013**). $CFHR3_{12}$ was prepared in an identical manner to $CFH_{67}$ with substitution of the final size exclusion chromatography step for purification using heparin affinity (Heparin FF column, GE Healthcare, buffer A: 50 mM Tris pH 7.5, 10 mM NaCl, buffer B: 50 mM Tris pH7.5, 1 M NaCl, elution gradient: 50% over 15 CV).

Recombinant His-tagged CFHR3 was expressed in HEK293 cells using Lipofectmine 2000 (Invitrogen) as per manufacturer's instructions. Recombinant His-tagged CFHR3 supernatant was purified by a single affinity chromatography step using a His-Trap HP (GE Healthcare) as per manufacturer's instructions. The bound protein was eluted with 500 mM Imidazole and then dialysed against 10 mM sodium phosphate pH7.8.

CFHR4 was expressed transiently in cell line HEK293S GntI$^{-/-}$ from vector pEF-BOS. Transfection was performed using 293Fectin (Life Technologies, Carlsbad, CA) as per the manufacturers instructions. Cells were cultured in suspension using Freestyle 293 Expression medium (Life Technologies) at 37°C with 10% $CO_2$ for 96 hr prior to harvesting the media. CFHR4 was purified from the media using a Ni-NTA column (Qiagen) as per the manufacturer's instructions. Purified protein was dialysed into phosphate buffered saline (PBS) prior to use.

## Surface plasmon resonance

All surface plasmon resonance measurements were made at 25°C using either a Biacore 3000 (GE Healthcare) or ProteOn XPR36 (BioRad, Hercules, CA) instrument with buffer HBS-EP (10 mM Hepes pH7.4, 150 mM NaCl, 3 mM EDTA, 0.05% surfactant-P20). Proteins were immobilised via primary amine coupling with a mock activated–deactivated reference channel included on each. Measurement of the affinity between CFHR3$_{12}$ and fHbp V3 was performed by immobilising approximately 1200 RU fHbp on the surface of a CM5 sensor chip (GE Healthcare). Dilution series of CFHR3$_{12}$ ranging from 12.5 nM–0.2 nM were flowed over the surface at 40 µl/min using KINJECT with a contact time of 300 s and dissociation time of 1200 s. The surface was regenerated with 10 mM glycine pH 3.0 between each injection. Curves were mock subtracted and fit with a 1:1 Langmuir model using BiaEvaluation (GE Healthcare).

Analysis of binding of CFHRs and CFHR fragments to fHbp V3 was performed using a CM5 chip with approximately 1700 RU fHbp immobilised. 100 µl of each analyte at a minimum concentration of 200 nM was flowed over the surface at 20 µl/min using the KINJECT command with a dissociation time of 400 s. The surface was regenerated using 10 mM glycine pH 3.0 between injections. Curves were mock subtracted using BiaEvaluation.

Measurement of the large panel of fHbp mutants was performed by immobilization of the proteins on ProteOn GLM sensor chips (BioRad). Increasing concentrations of CFHR3$_{12}$ or fH$_{67}$ were injected over the flow channels at 60 µl/min for 240 s and allowed to dissociate for 1200 s. The surface was regenerated with 10 mM glycine pH 3.0 between each injection. Curves were mock subtracted and fit with a 1:1 Langmuir model using ProteOn manager software (BioRad).

## Microscale thermophoresis

fHbp V3 was labelled using the RED-NHS labelling kit (NanoTemper Technologies, Munich, Germany) as per the manufacturer's instructions. Dilution series of CFHR3$_{12}$ in the range of 740 nM–23 pM were set up using HBS-EP. Each sample also contained 4 nM labelled fHbp. Thermophoresis was measured using a Monolith NT.115 instrument (NanoTemper Technologies) at 22°C utilising hydrophilic treated capillaries (NanoTemper Technologies), 80% LED power and 60% LED power. All data were measured in triplicate across three independent dilution series. Data were analysed using the signal from Thermophoresis + T-Jump (NT Analysis software version 1.5.41, NanoTemper Technologies).

## Haemolytic assays

The effect of CFHR3, CFHR3$_{12}$, CFHR3$_{45}$, CFHR4, and CFHR4$_{45}$ upon AP activation was assessed using haemolytic assays that were performed in an identical manner to that described in *Goicoechea de Jorge et al. (2013)*. All measurements were performed in triplicate and are presented as haemolysis relative to the level of lysis without addition of CFHR proteins (0%) and 100% lysis by $H_2O$. Haemolytic assays with CFH-deficient serum (CompTech, Tyler, TX) were performed with 5% serum.

## Bacterial strains and growth

*N. meningitidis* was grown on brain heart infusion (BHI, Oxoid) agar supplemented with 5% Levinthal's base (500 ml defibrinated horse blood autoclaved with 1 l BHI) overnight at 37°C in the presence of 5% $CO_2$ (*Lucidarme et al., 2009*; *Jongerius et al., 2013*). M1239Δ*fhbp* was constructed as previously (*Jongerius et al., 2013*) and isogenic strains were constructed by complementing the strains with fHbp V1.P1, V1.P14, V2.P22, or V3.P28 amplified using primers listed in *Table 1*. Polymerase chain reaction products were ligated into pGCC4 (*Mehr and Seifert, 1998*). Transformation of *N. meningitidis* strain M1239Δ*fhbp* was performed as described previously (*Exley et al., 2005*).

## Serum survival assay

NHS were obtained by collecting whole venous blood and allowing it to coagulate at room temperature for 60 min before centrifuging at 3000×g for 20 min at 4°C. Sera were heat inactivated at 56°C for 30 min prior to use. *N. meningitidis* was grown overnight on BHI agar, re-suspended in PBS then diluted to $1 \times 10^5$ CFU/ml in DMEM (Sigma). A total of $1 \times 10^4$ CFU of bacteria were pre-incubated with 1 μM CFHR3, 1 μM CFHR3 domains or PBS for 10 min prior to incubation with 5% human sera or 5% heat inactivated sera for 30 min at 37°C in the presence of $CO_2$. Bacterial survival was determined by plating onto BHI agar. Relative survival was calculated from samples containing no CFHR3 and incubated with heat-inactivated sera and expressed as percentage survival.

For the AP assays, bacteria were grown on BHI agar supplemented with 1 mM IPTG. Other complement pathways were inhibited by the addition of 5 mM $MgCl_2$/10 mM EGTA, and bacteria were incubated in 25% sera as above. Statistical significance was tested using the one-way ANOVA multiple comparisons as implemented in GraphPad Prism v.6.0 (GraphPad Software Inc.) to compare means ±S.E.M. using a $p < 0.01$ cutoff for significance.

## Monoclonal antibody production

Recombinant $CFHR3_{45}$ conjugated to KLH (Imject, Pierce) was used as an antigen. Female BALB/c mice were immunised with 100 μg of recombinant protein emulsified in TitreMax Gold (Sigma, UK) subcutaneously on day 1, then with 100 μg of $CFHR3_{45}$ in PBS intraperitoneally (IP) on days 21 and 42. A booster of 200 μg of recombinant protein in PBS was given IP 3 days prior to cell fusion on day 63. NS0 myeloma cells were grown in RPMI 1640 (Lonza, UK) supplemented with 10% foetal bovine serum (FBS, Sigma, UK), 2 mM L-glutamine, penicillin (100 U/ml), and streptomycin (100 μg/ml) (*Galfre and Milstein, 1981*). Splenectomy was performed on day 66, and splenocytes fused to NS0 myeloma in polyethylene glycol (PEG 1500, Roche, UK) (*Mason et al., 1983*). Fused cells were then plated into 24-well plates in RPMI 1640 containing 2 mM L-glutamine, penicillin (100 U/ml), streptomycin (100 μg/ml), and 1% Ultroser G (Pal France). After 24 hr, 2% hypoxanthine aminopterin thymidine (HAT, Life technologies) was added to the wells. Hybridoma clone supernatants were screened by ELISA against recombinant $CFHR3_{45}$. Positive clones were re-plated and grown for a further 5 days before screening again by ELISA against the $CFHR3_{45}$ and all five CFHR proteins. Positive hybridoma cells were sub-cultured into 96-well plates at a concentration of 1 cell per well in RPMI 1640 containing 10% FBS, 2 mM L-glutamine, penicillin (100 U/ml), streptomycin (100 μg/ml), Ultroser G (1%), and BM Condimed H1 (1%, Roche, UK). After 5 days, wells containing single colonies were screened as before.

## Flow cytometry

Bacteria ($1 \times 10^9$ CFU/ml) were fixed in 1 ml of 3% formaldehyde for 2 hr then washed with PBS. To evaluate CFHR3 binding, $5 \times 10^7$ CFU/ml were re-suspended in 10 μl of NHS or 10 μl of sera from an individual with ΔCFHR3/1 for 30 min at room temperature. After two washes in 0.05% BSA/PBS,

**Table 1.** Primers used to generate bacterial strains

| Primer name | Strain amplified | Ref | Primer Sequence (restriction site underlined) |
|---|---|---|---|
| pgcc4V1.1 F | H44/76 | (*Jongerius et al., 2013*) | CGGTTAATTAAGGAGTAATTTTTGTGAATCGAACTGCCTTCTGCT |
| pgcc4V1.1 R | H44/76 | (*Jongerius et al., 2013*) | CGGTTAATTAATTATTGCTTGGCGGC |
| pgcc4V1.14 F | NZ98/254 | | CGGTTAATTAAGGAGTAATTTTTGTGAACCGAACTGCC |
| pgcc4V1.14 R | NZ98/254 | | CGGTTAATTAATTATTGCTTGGCGGCAAGAC |
| pgcc4V2.22F | FAM18 | | CGGTTAATTAAGGAGTAATTTTTGTGAACCGAACTGCCTTCTGCT |
| pgcc4V2.22R | FAM18 | | CGGTTAATTAACTACTGTTTGCCGGCGATGC |
| pgcc4V3.28 F | M1239 | | CGGTTAATTAAGGAGTAATTTTTGTGAATCGAACTGCCTTCTGCT |
| pgcc4V3.28 R | M1239 | | CGGTTAATTAACTACTGTTTGCCGGCGATGC |

binding was detected following incubation with the anti-CFHR3 mAb (HSL1 this study) for 30 min at 4°C in 50 μl of PBS. Cells were washed twice in 0.05% BSA, then resuspended in 50 μl of goat anti-mouse IgG-Alexa Fluor 647 conjugate (1 in 1000 dilution in PBS; Molecular Probes, Life Technologies) and incubated for 30 min at 4°C. Samples were run on a FACSCalibur (BD Biosciences), and at least $10^4$ events recorded before results were analysed by calculating the geometric mean FL-4 in FlowJo vX software (Tree Star).

To evaluate the influence of CFHR3 on CFH binding to bacteria, $5 \times 10^7$ CFU/ml of *N. meningitidis* M1239 were incubated in 10 μl of *ΔCFHR3/1* sera and twofold dilutions of recombinant full-length CFHR3 (from 1 μM) for 30 min at room temperature. CFHR3 binding was detected with the anti-CFHR3 mAb (HSL1 this study) or an anti-human CFH mAb (MRC OX24, 0.1 μg/ml) (*Sim et al., 1983*), followed by incubating with goat anti-mouse IgG-Alexa Fluor 647 conjugate (1 in 1000 dilution in PBS; Molecular Probes, Life Technologies) and analysed by flow cytometry. At least $10^4$ events were recorded before results were analysed by calculating the geometric mean FL-4 in FlowJo vX software (Tree Star). Maximum fluorescence for CFHR3 was normalised to the addition of 1 μM exogenous CFHR3, whereas the maximum fluorescence for CFH was normalised without the addition of exogenous CFHR3. Statistical significance was tested using the one-way ANOVA multiple comparisons as implemented in GraphPad Prism v.6.0 (GraphPad Software Inc.) to compare means ±S.E.M. using a $p < 0.01$ cutoff for significance.

## Acknowledgements

JJEC and RME are funded by Wellcome Senior Investigator Awards to SML (100298) and CMT (102908), respectively, PNW by the Oxford Martin School Vaccine Design Institute, HL by an MRC programme grant G0900888 to CMT and SML. MCP is a Wellcome Trust Senior Fellow in Clinical Science (WT082291MA) and THM is funded by this fellowship. EGdeJ is funded by an Imperial College Research Fellowship. We thank Nicola Ruivo for technical assistance. Work in CMT's lab is funded by Wellcome Trust Investigator award (WT102908MA), a Programme grant from the MRC (G0900888), The Oxford Martin School and EUCLIDS (FP7 GA no. 279185).

## Additional information

### Funding

| Funder | Grant reference number | Author |
|---|---|---|
| Wellcome Trust | 100298 | Joseph JE Caesar, Susan M Lea |
| Wellcome Trust | 102908 | Christoph M Tang |
| Oxford Martin School, University of Oxford | Vaccine Design | Philip N Ward, Christoph M Tang, Susan M Lea |
| Medical Research Council | G0900888 | Hayley Lavender, Christoph M Tang, Susan M Lea |
| Wellcome Trust | 082291 | Talat H Malik, Matthew C Pickering |
| Imperial College London | Research Fellowship | Elena Goiecoechea De Jorge |
| European Commission | EUCLIDS FP7 GA no. 279185 | Christoph M Tang |

The funders had no role in study design, data collection and interpretation, or the decision to submit the work for publication.

### Author contributions

JJEC, CMT, SML, Conception and design, Acquisition of data, Analysis and interpretation of data, Drafting or revising the article; HL, PNW, JE, EC, Acquisition of data, Analysis and interpretation of data, Drafting or revising the article; RME, Constructed the natural fHbp variants; THM, Provided full-length CFHR3 and ΔCFHR-3/1 sera; EGDJ, Drafting or revising the article, Contributed unpublished

essential data or reagents; MCP, Analysis and interpretation of data, Contributed unpublished essential data or reagents

### Ethics

Human subjects: Tissue samples (in this study sera) were provided by the Imperial College Healthcare NHS Trust Tissue Bank (ICHTB). Other investigators may have received samples from these same tissues. The research was supported by the National Institute for Health Research (NIHR) Biomedical Research Centre based at Imperial College Healthcare NHS Trust and Imperial College London. The views expressed are those of the author(s) and not necessarily those of the NHS, the NIHR or the Department of Health. Volunteers were provided with the study information sheet (Donating Biological Samples for Research - Information for Donors, www1.imperial.ac.uk/tissuebank/tissuecollection/healthyvolunteers/) and informed consent obtained. ICHTB is approved by National Research Ethics Service to release human material for research (12/WA/0196).

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
