## [Decision Letter]

Thank you for sending your work entitled “Competition between antagonistic complement factors for a single protein on *N. meningitidis* rules disease susceptibility” for consideration at *eLife*. Your article has been favorably evaluated by Tadatsugu Taniguchi (Senior editor) and 3 reviewers, one of whom is a member of our Board of Reviewing Editors.

The Reviewing editor and the other reviewers discussed their comments before reaching the decision, and the Reviewing editor has assembled the following comments to help you prepare a revised submission.

This interesting and clinically relevant study provides data that allow a plausible explanation of human variability in outcomes of *Neisseria meningitidis* infection by defining both bacterial and host components that can vary to determine whether complement-mediated killing occurs within human serum. Previously, it is known that fHbp, a surface exposed lipoprotein from *N. meningitidis*, can bind to complement factor H (CFH), playing an important role of evasion of complement-mediated killing of the bacteria. Because of this, study of fHbp binding to CFH has important implications in vaccine design and improvement. In this work, the authors pursue a result from a genome-wide association study (GWAS) study of *N. meningitidis* that implicates human variability in CFHR3 as associated with host susceptibility. They demonstrate that CFHR3 could also bind to fHbp with a similar affinity as CFH and went further to show that the binding surface on fHbp by CFH and CFHR3 completely overlaps. They then demonstrate convincingly that CFHR3 promotes complement activation by competing with CFH on the bacterial surface. Finally, they further propose a model that the relative levels of two complement and related proteins (CFH and CFHR3) in the circulation system is a critical factor for determining the host susceptibility to meningococcal disease. In general, the study is very clearly written and provides an exciting and generally interesting observation. The concept proposed is novel and highly interesting given that there are five CFHR proteins that are often involved in binding to surface proteins of various bacterial pathogens. The paper is potentially suitable for *eLife* after addressing the following points.

1) As a rationale for the study and also a clinically relevant conclusion, the authors point to a recent GWAS study showing linkage disequilibrium to a specific single nucleotide polymorphism (SNP) and hence indicating the possibility that the linkage could be a result of deletion of CFHR3/CFHR1. In reading the Nature Genetics report, though, it does not appear that the CFHR3/CFHR1 deletion itself is linked to disease risk, but the association is rather to the CFHR3 SNP that is outside of the deleted region. The authors should clarify to readers the exact understanding of the relationship between *N. meningitidis* risk and the CFHR3/CFHR1 deletion. If that connection is not established through population studies then what could be the proposed mechanistic link to the CFHR3 polymorphism(s)?

2) In contrast to the binding assays, the actual differences in relative survival of *N. meningitidis* in the presence and absence of CFHR3 are rather modest (Figures 3 and 4). In addition, only one serum concentration and one protein concentration is used. In what is likely to be optimized conditions, the differences are <two-fold. Given the non-linear activities of the complement alternative pathway (AP) and amplification loop, it is not certain how modest changes in serum concentration would impact the results with different CFHR3/CFH ratios. To address this issue, the authors should present results of killing assays using a range of serum and protein concentrations. The authors should also include the fHbp-deficient *N. meningitidis* mutant strain as well as complementation by E255A as controls. Moreover, statistics should be also performed on the differences between normal human serum (NHS) and heat-inactivated serum with different CFHR3 proteins with p values calculated.

3) Serum killing assays are of course only an approximation of the complex bacterial protective response in vivo. Although there are no good animal models of human CFHR activities, inclusion of human neutrophils can provide a more complete analysis of the relationship between serum complement-bactericidal activity and bacterial killing (see for instance Ross et al, J Infect Dis. (1987) 155 (6): 1266-1275). The authors should perform such combined serum- and cell-based assays in which a more complete role of complement, and in particular its C3 fragment receptors on neutrophils, can be evaluated for their contribution to bacterial killing.

4) CFH and CFHR3 are generally considered to have similar function of negatively regulating the complement activity. However, the authors indicate here and in previous studies that this perception is wrong and CFHR3 can compete with CFH to bind the C3b complex and therefore counteract the negative function of CFH. The authors should go beyond the erythrocyte haemolysis assay in Figure 1 and examine the general function of CFHR3 in serum-mediating killing of bacteria that have nothing to do with CFH/CFHR3 binding (also consider using their CFHR3-deficient sera). This is particularly important in that pathogens exhibit a wide variety of mechanisms to evade the in vivo immune response, most of which do not directly target the complement system. In that context, additional support for an important role by this mechanism would add substantially to the relevance and potential impact of the study.

5) According to the authors' model, the function of CFHR3 should depend on the presence of CFH. However, there were no direct evidences demonstrating this point. Can CFH-deficient serum be found? If not, is it possible to deplete or block CFH by using CFH-specific antibody? In the latter, the authors can then examine whether exogenous CFHR3 still has any effect on complement-mediated killing in the sera in the absence of CFH function.

---

## [Author Response]

1) As a rationale for the study and also a clinically relevant conclusion, the authors point to a recent GWAS study showing linkage disequilibrium to a specific SNP and hence indicating the possibility that the linkage could be a result of deletion of CFHR3/CFHR1. In reading the Nature Genetics report, though, it does not appear that the CFHR3/CFHR1 deletion itself is linked to disease risk, but the association is rather to the CFHR3 SNP that is outside of the deleted region. The authors should clarify to readers the exact understanding of the relationship between N. meningitidis risk and the CFHR3/CFHR1 deletion. If that connection is not established through population studies, then what could be the proposed mechanistic link to the CFHR3 polymorphism(s)?

The GWAS study by Davila *et al.* did not show an association with the CFHR3‐1 deletion polymorphism as evident by the lack of an association between macular degeneration and the typed rs6677604 SNP (Davila et al., Nature Genetics 2010, Supplemental data Table 3). This SNP tags the CFHR3–1 deletion in many populations. The statistically significant associations in the GWAS study consisted of SNPs within CFH, within CFHR3 and within CFHR1, although the latter failed genotyping in one of the cohorts [Spanish] so was not included in the combined analysis (Davila et al., Nature Genetics 2010, Table 2). On the data presented it was not possible to conclude that the identified SNPs were causal because many of them were in linkage disequilibrium across the CFH–CFHR locus. In fact Davila et al. make the point “these CFH and CFHR SNPs are statistically indistinguishable from one another, and further work, such as deep resequencing, may be needed to identify the causal variant or variants within the associated region if the currently identified SNPs are not the functional variants”. What we can robustly conclude from the existing data is that variation within the CFH–CFHR locus influences susceptibility to macular degeneration. This includes variation within CFH (not unexpected from observations from complement deficiency states) and within the CFHR protein family (unexpected). Our functional studies support an association with variation within CFHR3. We have modified the text to reflect these comments.

*2) In contrast to the binding assays, the actual differences in relative survival of N. meningitidis in the presence and absence of CFHR3 are rather modest (*Figures 3 and 4*)*.

The difference in survival on the addition of CFHR3 is approximately 40% in the assay we performed that was over the course of 30 minutes. Although the difference appears modest to the reviewer, this will be marked during bacteraemia over hours in an infected individual. For example, over two hours the presence of CFHR3 would result in 88% less in survival of bacteria in the presence of CRHR3 compared to an individual with ΔCFHR3‐CFHR1 (and a 95% difference over 3 hrs).

*In addition, only one serum concentration and one protein concentration is used. In what is likely to be optimized conditions, the differences are <two-fold. Given the non-linear activities of the AP and amplification loop, it is not certain how modest changes in serum concentration would impact the results with different CFHR3/CFH ratios. To address this issue, the authors should present results of killing assays using a range of serum and protein concentrations. The authors should also include the fHbp-deficient N. meningitidis mutant strain as well as complementation by E255A as controls*.

As suggested by the reviewers, we have evaluated the effect of the different concentrations of CFHR3 on the survival of wild‐type bacteria and the isogenic fHbp mutant. The amount of serum in both the NHS and AP assays was kept constant to give approximately 80% bacterial survival without additional CFHR3; this enables the impact of this de‐regulator (which was expected to decrease survival) to be determined. The results demonstrate that the effect of CFHR3 (Mol Microbiol. 2011 Sep;81(5):1330-42; Nat Rev Microbiol. 2012 Jan 31;10(3):178-90) on meningococcal survival is dose and fHbp dependent (Figure 1). We now include these results as supplementary data (Figure 3—figure supplement 2, panel a).

*Moreover, statistics should be also performed on the differences between NHS and heat inactivated serum with different CFHR3 proteins with p values calculated*.

We have added the relevant statistical analyses to the Figure (Figure 3 in our revised manuscript) as requested.

*3) Serum killing assays are of course only an approximation of the complex bacterial protective response in vivo. Although there are no good animal models of human CFHR activities, inclusion of human neutrophils can provide a more complete analysis of the relationship between serum complement-bactericidal activity and bacterial killing (see for instance Ross et al, J Infect Dis. (1987) 155 (6): 1266-1275). The authors should perform such combined serum- and cell-based assays in which a more complete role of complement and in particular its C3 fragment receptors on neutrophils can be evaluated for their contribution to bacterial killing*.

This is an interesting suggestion. Marked susceptibility to meningococcal disease is seen in individuals with rare complement defects (especially loss of components of the terminal pathway, which is necessary for bacteriolysis and not opsonphagocytosis), those receiving C5 inhibitors (i.e. eculizumab), and inherited defects of the AP (i.e. properdin and factor D deficiencies). Furthermore, the established correlate of protection against meningococcal disease is the presence in the serum of bactericidal antibodies (i.e. antibodies that trigger the classical complement pathway and cause bacteriolysis).

In contrast, neutropaenic patients and people unable to generate a neutrophil anti-microbial oxidative burst do not exhibit high rates of meningococcal disease. This is consistent with the detection of diplococci in neutrophils in the cerebrospinal fluid and blood of patients with meningococcal disease (the bacterium was originally named *Diplococcus intracellulare*) and the resistance of pathogenic Neisseria to killing by neutrophils in the laboratory (shown by us and others, e.g., Mol Microbiol. 2011 Sep;81(5):1330-42; Nat Rev Microbiol. 2012 Jan 31;10(3):178-90). Work cited by the reviewer used a large number of uncharacterised bacterial strains and no details were provided about the method of recovery of neutrophils (which affects their activation status/NET formation).

Therefore neither genetic, clinical nor laboratory evidence implicates polymorphonuclear leukocytes and opsonphagocytosis in innate immunity against the meningococcus. Therefore work with polymorphonuclear leukocytes would not contribute any further insights into the effects of CFHRs. However, in line with the reviewers’ comments, we now discuss this point in the manuscript.

*4) CFH and CFHR3 are generally considered to have similar function of negatively regulating the complement activity. However, the authors indicate here and in previous studies that this perception is wrong and CFHR3 can compete with CFH to bind to C3b complex and therefore counteract the negative function of CFH. The authors should go beyond the erythrocyte haemolysis assay in*
Figure 1
*and examine the general function of CFHR3 in serum-mediating killing of bacteria that have nothing to do with CFH/CFHR3 binding (also consider using their CFHR3-deficient sera). This is particularly important in that pathogens exhibit a wide variety of mechanisms to evade the in vivo immune response, most of which do not directly target the complement system. In that context, additional support for an important role by this mechanism would add substantially to the relevance and potential impact of the study*.

Our hypothesis, supported by the data in Figure 1, is that CFHR3 has no activity except in its ability to compete for the same binding sites as CFH. This is now further supported by the additional demonstration that CFHR3 *per se* has no effect on complement-mediated lysis in the absence of CFH (see Figure 1—figure supplement 1 and the answer to point 5 below). As suggested by the reviewer, to further exemplify this point we have now examined serum-mediated killing of *E. coli* since these do not express any CFH binding protein. These new data (now incorporated into Figure 3—figure supplement 2, panel b) clearly demonstrate that addition of CFHR3 has no direct effect on killing of this bacterium.

*5) According to the authors' model, the function of CFHR3 should depend on the presence of CFH. However, there were no direct evidences demonstrating this point. Can CFH-deficient serum be found? If not, is it possible to deplete or block CFH by using CFH-specific antibody? In the latter, the authors can then examine whether exogenous CFHR3 still has any effect on complement-mediated killing in the sera in the absence of CFH function*.

Working with CFH-deficient sera is difficult as it is prone to full complement activation following relatively minor insult. However, as suggested, we have now repeated the haemolytic assays in CFH-deficient sera and show that addition of CFHR3 has no effect on haemolysis in the absence of CFH (new Figure 1—figure supplement 1). Furthermore, reconstitution of the sera with CFH restores the ability of CFHR3 to promote haemolysis.